# Double-Glued Multi-Focal Bionic Compound Eye Camera

**DOI:** 10.3390/mi14081548

**Published:** 2023-07-31

**Authors:** Xin Feng, Xiao Lv, Junyu Dong, Yongshun Liu, Fengfeng Shu, Yihui Wu

**Affiliations:** 1State Key Laboratory of Applied Optics, Changchun Institute of Optics, Fine Mechanics and Physics, Chinese Academy of Sciences, Changchun 130033, China; fengxin20@mails.ucas.ac.cn (X.F.); 15764337115@163.com (X.L.); dongjunyu97@163.com (J.D.); yihuiwu@ciomp.ac.cn (Y.W.); 2Key Laboratory of Optical System Advanced Manufacturing Technology, Chinese Academy of Sciences, Changchun 130033, China; 3University of Chinese Academy of Sciences, Beijing 100049, China

**Keywords:** curved compound eye, ommatidium, multiple focal lengths, high resolution

## Abstract

Compound eye cameras are a vital component of bionics. Compound eye lenses are currently used in light field cameras, monitoring imaging, medical endoscopes, and other fields. However, the resolution of the compound eye lens is still low at the moment, which has an impact on the application scene. Photolithography and negative pressure molding were used to create a double-glued multi-focal bionic compound eye camera in this study. The compound eye camera has 83 microlenses, with ommatidium diameters ranging from 400 μm to 660 μm, and a 92.3 degree field-of-view angle. The double-gluing structure significantly improves the optical performance of the compound eye lens, and the spatial resolution of the ommatidium is 57.00 lp mm^−1^. Additionally, the measurement of speed is investigated. This double-glue compound eye camera has numerous potential applications in the military, machine vision, and other fields.

## 1. Introduction

Compound eyes have a wide field of view, low aberration, and high time resolution. The visual range of an insect’s compound eye can reach 180 degrees, and insect [1,2,3] imaging is distortion-free. It also has a very high sensitivity to moving objects and can quickly identify and locate moving objects, with a response speed that is more than five times that of humans. Researchers conducted research on artificial compound eyes after being inspired by the compound eyes of animals. The resolution of the prepared compound eye imaging system has been steadily increasing as scientific cognition and processing technology [4,5,6,7,8,9,10,11,12,13,14,15] advance. At the moment, artificial compound eyes are facing numerous challenges, and researchers are constantly investigating new technologies and materials in order to apply artificial compound eyes to more scenes [16,17,18,19,20,21,22,23] and solve problems that traditional optical systems cannot. The current research trend focuses on high-resolution compound eye lenses [24,25,26].

Many methods for preparing compound eye lenses have been developed in recent years, including photoresist hot melting, ultra-precision machining, laser direct writing, non-contact hot pressing, and inkjet printing, among others. The photoresist hot melting method [27,28,29,30,31] involves melting a photoresist column and then forming a microlens under the influence of liquid surface tension. At the moment, one of the most common methods for preparing compound eye lenses is the hot-melt photoresist method. This method is simple to use and requires little preparation; however, the height of the ommatidium cannot be precisely adjusted, and the diameter of the ommatidium cannot be made very large. The surface of the ommatidium will sag as a result of the contact angle, reducing the imaging quality. Ultra-precision machining [32,33,34,35] is a technique for producing compound eye lenses using an ultra-precision machine tool and a single-point diamond as props. The workpiece processed by this method is difficult to assemble and adjust, the size of the tool will affect the surface accuracy, the processing cycle is long, the surface roughness of the compound eye is low, and the preparation cost is high, limiting its application. Laser direct [36,37] writing is a technique for directly processing a material substrate with a femtosecond laser. The exposure of the laser at different positions is controlled by a parametric design in order to process the compound eye lens. This method of preparation has the disadvantages of a lengthy preparation period, a high cost, and difficulties in terms of mass production. Non-contact hot pressing [38,39,40] involves covering a through-hole array mold with polymer material and applying a specific temperature and pressure to cause the polymer material to transition between vitrification and melting, resulting in the formation of microlenses in some materials in the through-holes. Ink-jet printing [41,42] is a technique for spraying microlens droplets onto a base material and then curing them with ultraviolet light to create a microlens array. This method is dependent on the tension of the liquid surface, so the surface roughness of the lens is good, but it is difficult to accurately control the ink jet quantity of each ommatidium, and the error is large.

Negative pressure forming and replication transfer technology has been shown to be an efficient and accurate compound eye lens preparation technology [43]. Negative pressure forming and replication transfer technology can transfer microlens arrays into curved surfaces. The preparation efficiency has been improved. Given the low resolution of compound eye lenses and the difficulty in directly integrating curved compound eyes with plane image sensors, this paper proposes a design method for preparing double-glued compound eye lenses on N-BK7 lenses that effectively reduces the influence of aberration and improves the imaging resolution. The compound eye lens prepared with NOA63 ultraviolet curing adhesive shortens the preparation time and has a good transmittance and clear imaging, allowing for the future large-scale production of high-definition compound eye cameras. The curing time is within 10 min, and the transmittance is above 96%.

## 2. Results and Discussion

### 2.1. Design and Manufacture of a Double-Glued Compound Eye Lens

Because of its unique application scene, the aperture of the compound eye cannot be very large, and preparing aspherical microlens under the current micro-nano manufacturing process is extremely difficult. The double-glued compound eye lens uses materials with different refractive indexes, effectively correcting the effects of spherical and chromatic aberration. Furthermore, the field of view of a single ommatidium of a compound eye lens is small, as is the influence of off-axis aberration. The double-glued method ensures ommatidium coaxiality, which improves the optical performance of the compound eye lens.

Negative pressure forming is used in the design of double-glued compound eye lenses, and ommatidiums are arranged in a ring shape step by step. The front end of the double-glued compound eye lens is an N-BK7 lens, and the back end is an NOA63-cured microlens array that is directly connected with CMOS. The diameter of the used N-BK7 is 25.4 mm, the radius of curvature is 24.53 mm, the center thickness is 9 mm, and the edge thickness is 1.9 mm. The N-BK7 lens has a refractive index of 1.52 and an Abbe number of 64. NOA63 has a refractive index of 1.56 and an Abbe number of 43, according to official data. The color difference can be mitigated by combining these two materials. We ran simulations to compare the compound eye lens with and without double gluing. The RMS value of the compound eye lens with double gluing was 2.501, while the RMS value of the compound eye lens without double gluing was 3.517, representing a 28.9% relative increase. As shown in Figure 1, it is clear that the compound eye lens with double gluing can improve the imaging quality.

The compound eye lens’s ommatidiums are then designed. Because the focal length of each ommatidium gradually increases with the distance from the center, changing the diameter and height of each ommatidium is required to unify the focal length of the microlens on a plane so that all lenses can clearly image. The ommatidium designed in this study has six levels, with diameters from the center to the edge of 400 μm, 420 μm, 450 μm, 480 μm, 540 μm, and 660 μm, and numbers of 1, 6, 12, 18, 22, and 24. The height of the ommatidium is determined by the fact that the PDMS film is under negative pressure, the micropores have different diameters, and the film deformation is different. The distance between adjacent ommatidiums is 100 μm, which prevents imaging interference and gives the silicon wafer of the microporous array enough strength during the negative pressure forming process.

The silicon wafer used to etch the micro-hole array die has low deformation and high mechanical strength. The microporous array die for negative pressure is created using photolithography, back etching, ICP etching, and other technological methods, and the size of the through holes is determined using the above design. Because of its ductility, PDMS can be used as a film to cover the micro-hole array mold during the negative pressure forming process. The PDMS prepolymer solution and curing agent were mixed 10:1 in a vacuum box to remove bubbles caused by stirring. Pour the mixed PDMS liquid onto a flat plate, rotate it with a spin coater at 3000 r/min, and place the obtained PDMS film in an oven at 80 °C for 20 min to heat and cure before preparing the PDMS film with a thickness of about 10 μm.

Negative pressure is applied below the mold as the prepared PDMS film is covered on the micro-hole array mold. The position of the through hole of the film is depressed by atmospheric pressure, and the height of the depression is the height of the compound eye lens’s ommatidium. Drop NOA63 UV curing adhesive onto the top of the film, cure for 3 min with a UV curing lamp, and remove the cured NOA63 UV curing adhesive, namely the planar microlens array.

Fix the planar microlens array, apply PDMS to it, and bake it at 80 degrees for 3 h to cure the PDMS. The PDMS mold was removed from the plane microlens array after curing to create a PDMS mold with a concave hole array. The PDMS mold is pliable and can be used as a negative pressure mold to prepare double-glued compound eye lenses. Negative pressure is applied to the lower part of the PDMS mold to deform it into a curved surface while maintaining constant air pressure, and NOA63 ultraviolet curing glue is dripped again, taking care not to introduce bubbles. After dripping, the upper part is covered with an N-BK7 lens, while the center is kept, and ultraviolet curing is performed for 8 min before removing the lower PDMS mold to obtain a double-glued multi-focal bionic compound eye lens, as shown in Figure 2.

### 2.2. Surface Morphology of the Double-Glued Compound Eye Lens

Figure 3a depicts the surface morphology of a double-glued compound eye lens. Figure 3b shows a scanning electron microscope (SEM) photograph of the lens’s surface. The microlenses on the N-BK7 convex lens are evenly distributed. The ultra-depth microscope photos show that the sizes of each ommatidium of the prepared double-glued compound eye lens are 395.96 μm, 418.05 μm, 447.12 μm, 476.19 μm, 533.60 μm, and 654.54 μm, as shown in Figure 3c. Measure the height of each ommatidium with a step meter, as shown in Figure 3d. The overall size is slightly smaller than the design value, owing to the thickness of the PDMS film. Furthermore, shrinkage of NOA63 during ultraviolet curing is an important reason, which can be solved in the future by changing the design value and offsetting the error.

The focal length of the double-cemented compound eye lens’s ommatidium can be calculated using the following formula:(1)$$f=\frac{h^{2}+{\left(\frac{D}{2}\right)}^{2}}{2h\left(n-1\right)}$$
where d and h are the microlens’s diameter and height. The focal length of the double-glued compound eye lens’s ommatidium is calculated as shown in Figure 3e,f. All ommatidiums of the double-glued compound eye lens have uniform optical properties due to the good surface and technological process.

### 2.3. Characterization of the Optical Performance of the Double-Glued Compound Eye Lens

A focus measuring device is used to demonstrate the imaging ability of the double-glued compound eye lens. Figure 4a depicts a focus measuring device. A CMOS, objective lens, optical filter, double-glued compound eye lens, beam expanding collimator, and 632 nm laser are shown from left to right. The laser excites the light source in this experiment, and a collimated laser covering all microlens arrays is formed by using the beam-expanding collimating device. The collimated laser is attenuated by the attenuator after passing through the double-glued compound eye lens, amplified by an objective lens, and captured by CMOS. The figure depicts the shooting outcome. The spot size is uniform, the focus is on the same plane, and the lens’s focusing performance is excellent. As shown in Figure 4b, we put the double-glued compound eye lens in a camera and shoot the letter version to calculate its field of view. The camera is 12 cm away from the letter version, and the distance between the two furthest letters is 25 cm. The field-of-view angle of the double-glued compound eye lens can reach 92.3 degrees based on trigonometric function calculation.

The field-of-view angle can be increased by increasing the number of ommatidiums, but this requires CMOS reception of a larger target surface. The USAF1951 resolution test board is used to evaluate the double-glued compound eye lens. The sixth element of the fifth group can be clearly identified in Figure 5a, indicating that the resolution of the double-glued compound eye lens is 57.00 lp mm^−1^. Figure 5b depicts photos taken by Lena with the double-glued compound eye lens, and all of the microlenses can clearly image, which is comparable to the focus map results.

The imaging ability of the double-glued compound eye lens for the letter “A” was tested using an optical microscope. The image of the letter “A” can be clearly displayed without obvious distortion, as shown in Figure 5c, and the light transmittance is good, demonstrating the excellent optical performance of the double-glued compound eye lens.

### 2.4. Application of the Double-Glued Compound Eye Lens

Motion sensitivity is an important feature of the compound eye. This study achieved the measurement of car speed in motion using the Gaussian Mixture Model (GMM). GMM identifies the moving car as the foreground and divides the photographed image into foreground and background to enable additional speed measurement. As shown in Figure 6a, we set up a speed measuring device that secures the double-glued compound eye camera in front with a smooth track. The car is released at a certain height and slides down the track at a uniform speed by changing the inclination angle of the track. The trolley’s movement is measured, and the total length of the track is 60 cm, after which the trolley is released from the top. The process of the car sliding down from the top was captured on film. The image’s sampling rate was 5 fps, and the experiment’s sampling time was 0.8 s and 1.4 s, respectively. The car moves from the right to the left side at a constant speed. The images captured by the double-glued compound eye camera at 0 s, 0.8 s, and 1.4 s are shown in Figure 6b. The recognized cars are represented by the photos on the left, the GMM algorithm process is shown in the middle, with the white pattern as the judgment process, and the green box is shown on the right. The calculated average speed of the car is approximately 26 cm/s. It has been demonstrated that the device has a high potential for use in tracking moving objects.

## 3. Conclusions

The double-glued compound eye lens was created using lithography, soft lithography, and negative pressure forming technology. By using this method, the resolution of the compound eye lens is improved and the shape of the ommatidium can be precisely controlled. The prepared double-glued compound eye lens’s spherical surface contains 83 microlenses, achieving a field-of-view angle of 92.3 degrees, a double-glued structure, and a good surface quality, which improves the compound eye lens’s optical performance, with a spatial resolution of 57.00 lp mm^−1^. Currently, however, the resolution of compound eye lenses is still low, making it difficult to achieve clear imaging over long distances. The double-glued compound eye lens is capable of tracking and measuring the speed of moving objects in space. Experiments show that the double-glued compound eye lens has an excellent imaging performance over a wide field of view and can be connected directly to CMOS. We believe that this double-glued compound eye lens has a lot of potential for machine vision and other applications.

## Figures and Tables

**Figure 1 micromachines-14-01548-f001:**
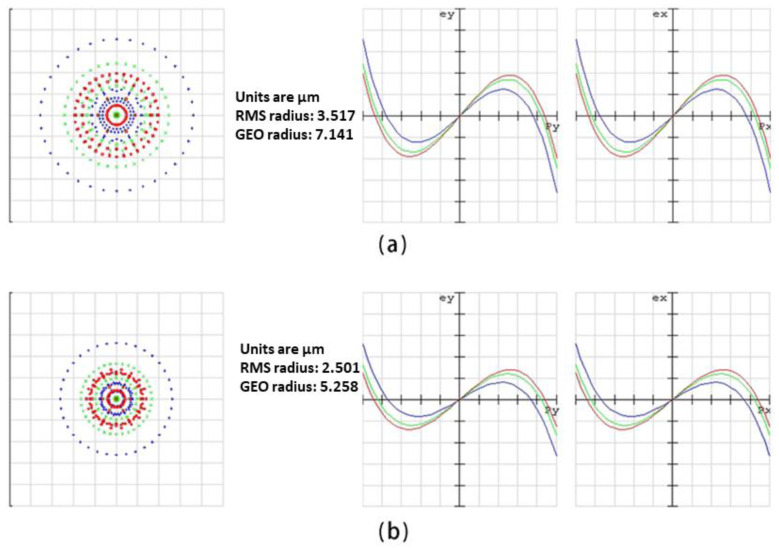
(**a**) The ray fan of the single compound eye lens. (**b**) The ray fan of the double-glued compound eye lens.

**Figure 2 micromachines-14-01548-f002:**
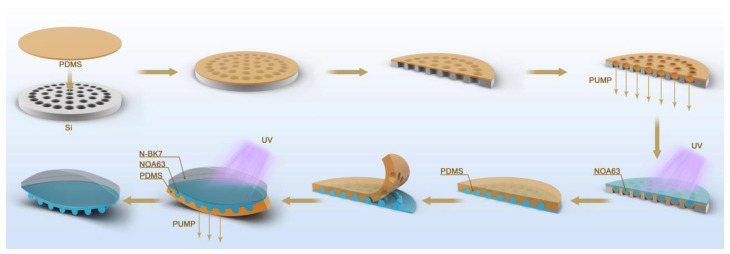
Double-glued compound eye lens preparation process.

**Figure 3 micromachines-14-01548-f003:**
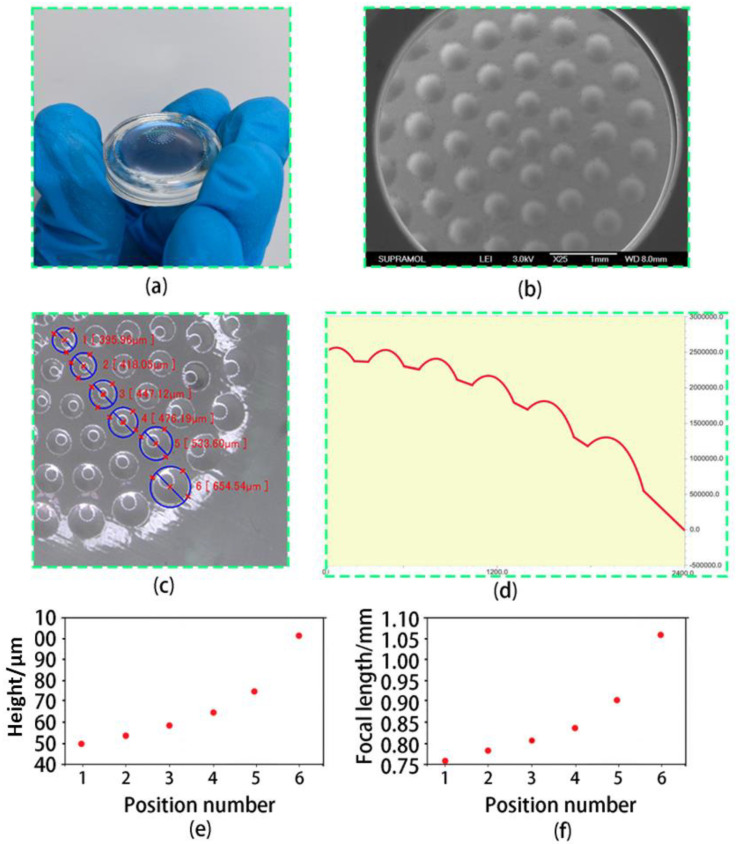
(**a**) A photo of the double-glued compound eye lens. (**b**) A photo taken with a scanning electron microscope. (**c**) Measurement of diameter of the double-glued compound eye lens. (**d**) Profiler measurements of the height of the compound eye. (**e**) The height of the compound eye lens. (**f**) The focal length of the compound eye lens.

**Figure 4 micromachines-14-01548-f004:**
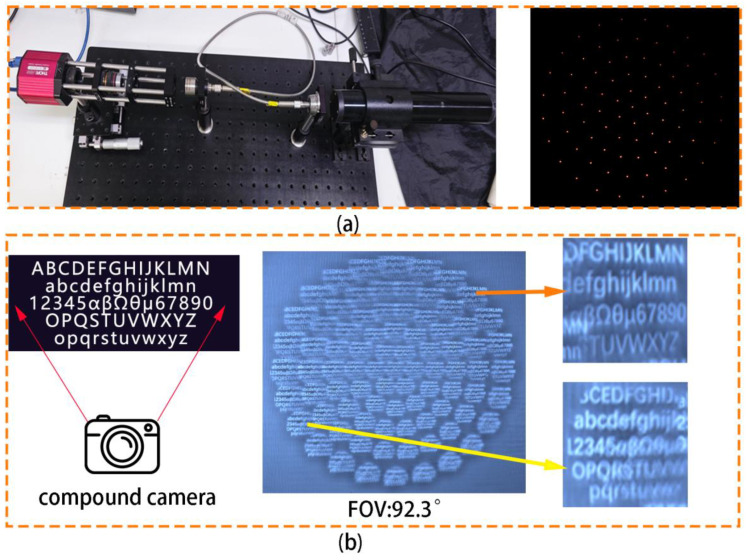
(**a**) The focal spot of each ring. (**b**) Field angle measurement of compound eye camera.

**Figure 5 micromachines-14-01548-f005:**
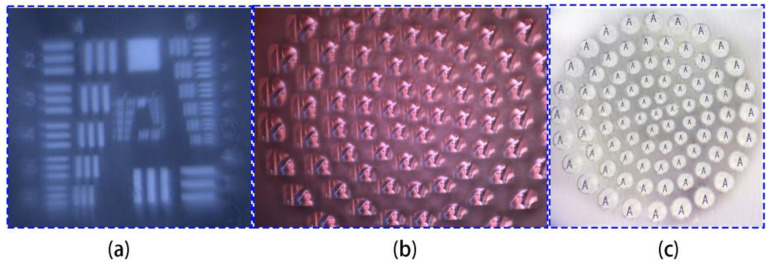
(**a**) Compound eye resolution test. (**b**) Shoot results of Lena. (**c**) Compound eye camera captures letter “A”.

**Figure 6 micromachines-14-01548-f006:**
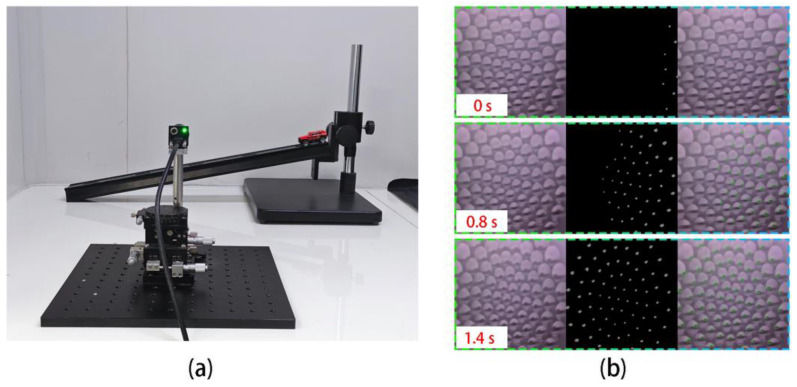
(**a**) Speed measurement device. (**b**) Measurement of the speed of a moving car using the compound eye camera.

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
