# Peer review of "Double-Glued Multi-Focal Bionic Compound Eye Camera"

_micromachines, 2023, doi:10.3390/mi14081548_

Round 1

Reviewer 1 Report

This paper introduces a method for creating a double-glued multi-focal bionic compound eye camera using photolithography and negative pressure molding technology. Some suggestions are as follows:

1. Some important references are missing in the Introduction, e.g., line 35, "...The current research trend is for high-resolution compound eye lenses...", line 62. "...Negative pressure forming and replication transfer technology has been shown to be an efficient and accurate compound eye lens preparation technology...", and so on.

2. The Introduction describes the negative pressure forming and replication transfer technology and the main contributions of this paper too briefly.

3. The description of "The compound eye lens prepared with NOA63 ultraviolet curing adhesive shortens the preparation time and has good transmittance and clear imaging" in the Introduction lacks data support.

4. "The compound eye camera has 83 microlenses..." in the Abstract contradicts "The prepared double-glued compound eye lens's spherical surface contains 87 microlenses..." in the Conclusions.

5. In line 200, "The images captured by the double-glued compound eye camera at 0s, 0s, and 1.4s are shown in Figure 6b...", the second 0s should be 0.8s.

6. Surface roughness and consistency data for the double-glued compound eye lens need to be presented.

7. The disadvantages and difficulties of the double-glued compound eye lens preparation method need to be introduced in the paper.

Author Response

Response to Reviewer 1 Comments

Thank you for your precious comments and advice. Those comments are all valuable and very helpful for revising and improving our paper, as well as the important guiding significance to our researches. We have studied comments carefully and have made correction which we hope meet with approval. The main corrections in the paper and the responds to the comments are as flowing:

Point 1: Some important references are missing in the Introduction, e.g., line 35, "...The current research trend is for high-resolution compound eye lenses...", line 62. "...Negative pressure forming and replication transfer technology has been shown to be an efficient and accurate compound eye lens preparation technology...", and so on.

Response 1: New references have been added to the article.

Point 2: The Introduction describes the negative pressure forming and replication transfer technology and the main contributions of this paper too briefly.

Response 2: The introduction has been modified.

Point 3: The description of "The compound eye lens prepared with NOA63 ultraviolet curing adhesive shortens the preparation time and has good transmittance and clear imaging" in the Introduction lacks data support.

Response 3: The corresponding data has been added in the article.

The light transmittance given by the official of the material and our test results are shown below.

Tested with Lambda 850, its light transmittance is above 96%.

Point 4: "The compound eye camera has 83 microlenses..." in the Abstract contradicts "The prepared double-glued compound eye lens's spherical surface contains 87 microlenses..." in the Conclusions.

Response 4: The numbers have been corrected.

Point 5: In line 200, "The images captured by the double-glued compound eye camera at 0s, 0s, and 1.4s are shown in Figure 6b...", the second 0s should be 0.8s.

Response 5: The numbers have been corrected.

Point 6:Surface roughness and consistency data for the double-glued compound eye lens need to be presented.

Response 6: We measured three compound eye lenses using AFM, and their surface Ra values are 8.11nm, 6.64nm, and 7.49nm, as shown below.

Point 7:The disadvantages and difficulties of the double-glued compound eye lens preparation method need to be introduced in the paper.

Response 7:The disadvantages and difficulties of the double-glued compound eye lens have been added to the conclusion section of the article.

We sincerely hope that this revised manuscript has addressed all your comments and suggestions. We appreciated for reviewers’ warm work earnestly, and hope that the correction will meet with approval. Once again, thank you very much for your comments and suggestions. We would like to thank the referee again for taking the time to review our manuscript.

Reviewer 2 Report

In this manuscript a double-glued multi-focal bionic compound eye camera is proposed and fabricated by photolithography and negative pressure molding method. Its imaging performance has been investigated and it is demonstrated for measurement of speed of a moving toy car. The manuscript is well prepared and organized, So I would recommend publication of this manuscript after minor corrections. Please address the following comments:

1.       Typos: line 18, “lp mm-1” should be “lp mm-1; line 104 , “100 m” should be “100 μm”

2.       In Fig. 4(b), the letter picture showing on the left is different from what we observed in the compound lens.

3.       In the experiment of measuring the speed of the toy car. The car should accelerate on all its way on the track if it is freely sliding from the top to bottom, but the author mentioned that the car has a uniform speed. Do you take any actions to control the sliding speed of the car, if so why would you choose an angle titled track to do this experiment? Besides, the actual speed of the car should be calibrated and given in the manuscript and compare with that of the measured result using the compound camera.

Author Response

Response to Reviewer 1 Comments

Thank you for your precious comments and advice. Those comments are all valuable and very helpful for revising and improving our paper, as well as the important guiding significance to our researches. We have studied comments carefully and have made correction which we hope meet with approval. The main corrections in the paper and the responds to the comments are as flowing:

Point 1: Typos: line 18, “lp mm-1” should be “lp mm-1; line 104 , “100 m” should be “100 μm”

Response 1: The text has been corrected.

Point 2: In Fig. 4(b), the letter picture showing on the left is different from what we observed in the compound lens.

Response 2: The image has been modified as follows.

Point 3: In the experiment of measuring the speed of the toy car. The car should accelerate on all its way on the track if it is freely sliding from the top to bottom, but the author mentioned that the car has a uniform speed. Do you take any actions to control the sliding speed of the car, if so why would you choose an angle titled track to do this experiment? Besides, the actual speed of the car should be calibrated and given in the manuscript and compare with that of the measured result using the compound camera.

Response 3: We balance the gravity and friction of the toy car by controlling the angle of the track, and the toy car can slide down at a uniform speed. Due to the lack of precise speed measurement equipment in the laboratory and the relatively slow speed of the car, we are unable to accurately measure the speed of the car. In the following plan, we plan to measure the moving car, with the car moving at a constant speed, so that the actual value can be compared with the measured value.

We sincerely hope that this revised manuscript has addressed all your comments and suggestions. We appreciated for reviewers’ warm work earnestly, and hope that the correction will meet with approval. Once again, thank you very much for your comments and suggestions. We would like to thank the referee again for taking the time to review our manuscript.
